# Analyzing first aid in textbooks used by non-medical and paramedical students in Nepal: A need of further attention for snakebite management!

Deb Prasad Pandey[1,2*], Bishnu Prasad Khanal[3], Hardik Sapkota[2]

**1** Department of Veterinary Microbiology and Parasitology, Faculty of Animal Science, Veterinary Science and Fisheries, Agriculture and Forestry University, Rampur, Chitwan, Bagmati Province, Nepal, **2** Institute for Social and Environmental Research-Nepal (ISER-N), Fulbari, Chitwan, Bagmati Province, Nepal, **3** Independent researcher, Calgary, Canada

* debpandey@gmail.com

## Abstract

### Background

Despite the remarkable burden of snakebite envenoming (SBE) and placement of public health importance of SBE at global and national levels, a huge gap still exists in the first aid of snakebites. Herein, we aimed to evaluate the currently used textbooks to know improvements in advising standard first aid of snakebites in textbooks used in Nepalese schools and universities after a similar study published in 2013.

### Methodology/Principal Findings

We evaluated 46 recently edited textbooks used for teaching safety and first aid in Nepal during January–April 2024 involving thematic analytical approach. We performed an analytical review of safety and first aid in textbook. The counts of harmful or useless advice for first aid significantly decreased [p < 0.001] and standard first aid significantly increased [p < 0.001]. But, the proportions of advice for emergency transport, and going to an appropriate healthcare facility provided with anti-snake venom (RAFA) negligibly increased [p = 0.367 (right)]. However, 6–75% errors reflected the persistent use of teaching materials containing non-recommended first aid measures.

### Conclusions/Significance

Therefore, it is essential to update these textbooks including evidence-based, standard first aid for snakebites to increase the advice for appropriate care of snakebites at pre-hospital condition. We suggest authors revising those textbooks including the advice for application of pressure-immobilization bandaging (PIB) and local compression-pad immobilization (LCPI) by trained first aider and other first aid that can be applied by general people. PIB delays the onset of systemic neurotoxic

**Data availability statement:** All data are in the manuscript and Supporting Information files.

**Funding:** The author(s) received no specific funding for this work.

**Competing interests:** The authors have declared that no competing interests exist.

venom effects due to krait bites. LCPI delays systemic venom effects and minimize local toxicity (i.e., destruction of tissue at the site of the bite) due to cobra and all viperid snake venoms.

## Author summary

Snakebite envenoming is a public health important issue at global and national levels. However, there is a huge gap in the first aid of snakebites. Herein, during January–April 2024, we evaluated the recently edited, currently used 46 textbooks to know improvements in advising standard first aid of snakebites in textbooks used for teaching safety and first aid in Nepal after a similar study published in 2013. The counts of harmful or useless advice for first aid significantly decreased and standard first aid significantly increased. But, the proportions of advice for emergency transport, and going to an appropriate healthcare facility provided with anti-snake venom negligibly increased. 6–75% errors contained in those books reflected the persistent teaching of non-recommended first aid measures. Therefore, it is essential to update these textbooks including evidence-based, standard first aid for snakebites to increase the advice for appropriate care of snakebites at pre-hospital condition.

## Introduction

Snakebite envenoming (SBE) is a neglected public health problem in the tropics and the sub-tropics where snakes and humans encounter occasionally [1] resulting in snakebites affecting at least 2.7 million people annually [2]. Despite the placement of public health importance of SBE at global [3–6] and national levels [7–11], a huge gap still exists in utilizing standard first aid practices [12,13]. In agrarian and remote tropical and sub-tropical communities [1] the recommended first aid including the adjunct pharmacological interventions [14] increase health benefits [15] and decrease the treatment costs and deaths [15,16]. These practices are the standard first aid which is a set of actions performed immediately after snakebite, before the patient is carried to a snakebite treatment center (STC) to slow down the systemic venom effects. Snakebite victims adopt various first aid measures to delay or cease the venom effects until they are carried to the hospital for the definitive treatment [1,17–20]. All these measures (defined and listed in the Box 1) are considered recommended advice (RA). But, patients in these hotspots of snakebites commonly use non-standard first aid [12,19,21].

Rapid responses for effective first aid and early access to in-hospital treatment is important in the first few hours after a snakebite [22]. However, the lack of awareness among Nepalese students, teachers, and community people about the standard first aid and their inadequate ability to respond to snakebite [1,23,24] hinders public preparedness for the first aid and safety. In many cases with poor accessibility to health

care in Nepal, 41–68% snakebite patients [20,25] still rely mostly on substandard, unsafe, useless or non-recommended first aid before accessing a STC [54–58% [20,25] applied a tourniquet; 10–57% [19,20,25–27] approached traditional snakebite healers (and received incision of the bite site, suction of wound, ringing/deepening in potash solution, herbal medicine, and other traditional topical concoctions); 18% (n = 9) [28] of the traditional healers from Nepal mentioned that patients consulted the healers in parallel with the healthcare system; 1–5% [20,25] ingested chillies and topical used herbal medicine and honey believing to combat envenoming effects and squeezed the wound believing to ooze out venom with blood; 5% [25] applied snake stones; 3% [25] made local incisions to let out the injected venom; 8–90% [19,20,25] envenomed patients accessed a healthcare center supplied with no antivenom or no specialty or trained medical personnel for snakebite management], and many of them do not reach healthcare facilities [19,27,28]. Adoption of these traditional and non-recommended first aid is either harmful or useless and delays time (up to median of 3.3 hours [19]) to receive definitive medical care of envenoming in Nepal. This poses patients at the risk of morbidity or mortality [19,20,28–30]. All these substandard, unsafe, or useless measures (defined and listed in the Box 1) are considered non-recommended advice (NRA), herein. Moreover, suggestion to "carry the victims right away to a doctor or a hospital/ health center/health post" without disclosing the availability of antivenom is partially accurate (PAA) in envenomed victims requiring antivenom therapy. Visiting healthcare facilities supplied with no antivenom can be a cause of delay for proper treatment of snakebite. To minimize these consequences due to snakebites and meet with the goal of the World Health Organization (WHO) for halving global snakebite burden by the year 2030 [3], there is an urgent need for increased awareness on effective first aid in Nepal. This understanding can be achieved through the teaching of proper first aid and informing standard practices to visitors and inhabitants of snakebite prone regions.

The first aid has important role to improve the hospital management and outcomes of snakebites [15,31] due to common medically relevant snake species of Nepal [i.e., *Naja naja*, *N. kaouthia*, *Bungarus caeruleus*, *B. lividus*, *B. walli*, *B. niger*, *Daboia russelii*, *Trimeresurus* spp., and *Ovophis monticola* [32]] because proper first aid given to snakebite victims [33,34] positively influence outcomes of in-hospital treatment of snakebites. Therefore, in addition to the readiness for in-hospital care of snakebites, an adequate understanding of appropriate first aid for snakebite is crucial/ urgent for the human populations inhabiting 48–50 districts of the lowlands and middle to higher hills of Nepal where people are at the risk of SBE [19,35] while sleeping at home (25–67%) or agricultural activities performed indoors or at crop fields (15–32%) [19,20,25,27], with 11% of collective envenoming rate [35]. These envenomings result in a minimum of 101 deaths annually [35]. Despite the noticeable impact of SBE [16,19,35], educational materials used in Nepalese schools and universities are insufficient to educate the vulnerable people because of inclusion of several non-recommended first aid in books [36] and lack of update on these textbooks still. This lacking might confuse teachers, students, and community people to accept the measures recommended by the WHO [31]. The Curriculum Development Center (CDC), Ministry of Education, Nepal Government, has not evaluated limitations of first aid mentioned in the books used in Nepalese schools and universities yet. These poor and incomplete books promote the use of non-standard first aid in Nepal and can be a cause of persistently adherence of community people towards substandard or harmful first aid. Therefore, periodic evaluation of educational materials used in formal education of medical and non-medical students is essential in preventing additional harm to snakebite patients due to improper and untimely first aid.

In Nepal, basic (grades 1–8) and secondary (grades 9–12) education system aims formal education through publicly and privately funded schools and religious schools. In the academic year 2024/25, a total of 35,447 schools [community and religious schools: 27,298 (77.0%), institutional (aka private) schools: 8,149 (23.0%); cited in: https://edusanjal.com/ blog/education-levels-nepal/; accessed on: 18th June 2025] enrolled 7,091,000 students nationwide (cited in: https://www. nrb.org.np/contents/uploads/2022/07/Education.xlsx; accessed on: 18th June 2025]. Similarly, basic technical education is provided by the Council for Technical Education and Vocational Training (CTEVT). The total CTEVT students enrolled in Diploma/PCL (Proficiency Certificate Level) in Health/Nursing (i.e., Diploma in Pharmacy/PCL of Pharmacy, PCL of Nursing, Community Health Worker, Community Medicine Assistant, Health Assistant, Aayurvedic Health Assistant, Aayurvedic

Auxillary Health Worker] was 12,747 (cited in: Appendix 4, available at: https://ctevt.org.np/documents/ctevt-annual-report-2081). Next, the number of students enrolled yearly for curriculum of Bachelor's Degree in Health/Nursing [Bachelor of Pharmacy (B. Pharm) and Bachelor of Science in Nursing (BSc Nursing), Bachelor of Nursing Science (BNS), Bachelor of Public Health (BPH), Bachelor of Ayurvedic Medicine and Surgery (BAMS)] in different Nepalese universities was 3,041 (S1 Table). Nationwide, 500 institutions affiliated to TU currently offer Bachelor of Education (B.Ed.) in Nepal (https://edusanjal.com/course/bachelor-of-education-bed-tribhuvan-university/). There was primary methodology of teaching with physical books in Nepalese educational institutions although Information and Communication Technology (ICT) is increasingly recognized as a crucial tool for improving education at all levels, from basic to higher education in this country. Therefore, this research has potential significance and applicability in education sectors, too.

Herein, we hypothesized that proportions of advice to use RA increased and NRA, PAA, and total erroneous advice decreased in recently evaluated textbooks compared to those evaluated in the past [36]. Additionally, we hypothesized that the significantly large effect size in differences of proportions for RA, NRA, PAA, and total erroneous advice in those books. Further, we assess deviations from or absence of standard pre-hospital snakebite management in recently edited textbooks used in schools and universities of Nepal and suggest book writers, teachers, and students to adhere in the standard first aid [31]. This article provides new information about the persistent teaching of harmful interventions for first aid of snakebites in Nepal as the currently used textbooks in Nepal contain first aid with potential of deviations from or absence of standard first aid. This provides an opportunity for improving education in snakebite management, starting in professional circles in Nepal and other nations where evaluation of the precision of the measures for first aid of snakebites is expected.

## Methods

### Ethics statement

This article does not include human or animal research. So, institutional ethical approval was not applicable. Despite the absence of direct research with human or animal subjects, informants (book author and educators) provided verbal assent, even if formal ethical approval was not necessary, prior to their interviews. These interviews were concentrated to the recently used textbooks containing safety and first aid advising first aid and their nationwide representation.

### Data collection strategies

We conducted an analytical review of snakebite first aid in textbook during January–April 2024, for which we adopted research tools used in previous study [36] and thematic analysis. To identify the recently used textbooks containing safety and first aid unit advising first aid of snakebites, we used two criteria: following up of the books listed in the previous study [36] and interviewing key informants. The key informants included a health education related book writer, five teachers who used to teach first aid for snakebites and other accidental illness in university and schools, and an administrator and science teacher of a school], who were actively involved in teaching in Bharatpur Metropolitan City, Chitwan District and first aid trainers, too (S1 Fig). Following their advice and the listed books [36], we visited government- and private-run schools and Tribhuvan University affiliated campuses across Chitwan and Nawalpur Districts of Nepal and collected the most recently edited 46 textbooks related to health education and biology subjects (S2 Table) that were commonly and currently used by non-medical students of classes four to seven (9–12 years) and classes 11–12 (16–17 years) in schools, as recommended by the CDC and the Higher Secondary Education Board, Bhaktapur and of B.Ed. 4th year [21 years and beyond], as recommended by the Tribhuvan University (TU), Kathmandu (S2 Table, Table 1) for formal education of first aid of snakebites nationwide. These textbooks also represented paramedical books used by medical undergraduates (Proficiency Certificate Level, 16–17 years) and graduates (Bachelor's Degree level, aged 18 and beyond), as recommended by the CTEVT (https://ctevt.org.np/introduction), Sanothimi, Bhaktapur and some Nepalese universities

**Table 1. Clustering of extracted advice (from 16 textbooks) for first aid after a snakebite into three themes [RA: recommendable advice; NRA: non-recommendable advice; PAA: partially accurate advice].**

| Themes | Quotes/statements indicating advice for first aid of snakebite | Books (N = 16) | Classes using books | Secondary codes of books (Please, see the S2 Table for the further details) |
|---|---|---|---|---|
| *RA* | "**La**y the patient to **the rest** in a safe place and **reassure** keeping calm and still, **provide emotional support**" | 13 | 6, 7, B.Ed., Paramedical classes | 6.12; 7.19; 7.20; 7.15; 7.16; B.39; P.40; P.44; P.45; P.46; P.43; P.41; P.42 |
| | "**Wash the wound** with soap water, potassium permanganate water, sterile saline or water, **keep the wound dry** with clean swab or cloth, and **mark the bite site properly**" | 9 | 6, 7, B.Ed., Paramedical classes | 6.11; 6.12; 7.19; 7.20; 7.15; 7.18; B.39; P.44; P.43 |
| | "Try to **identify snakes** involved in bite but without capturing or killing it", "If the snake has been killed, take it to the hospital with the victim", try to confirm bite observing teeth marks on the skin" | 7 | 6, B.Ed., Paramedical classes | 6.12; B.39; P.45; P.46; P.43; P.41; P.42 |
| | "Victim should not be allowed to move to **avoid muscular contraction**" | 7 | 6, 7, B.Ed., Paramedical classes | 6.11; 7.17; 7.18; B.39; P.44; P.41; P.42 |
| | "Call for an ambulance or expert help" | 2 | 7, B.Ed. | 7.16; B.39 |
| | "Immobilize bitten part by using PIB (pressure immobilization bandaging) or LCPI (local compression-pad immobilization)" | 11 | 7, Paramedical classes | 7.15; B.39; 7.20; 7.19; P.40; P.44; P.45; P.46; P.43; P.41; P.42 |
| | "Immobilize the affected body part" | 5 | 6, Paramedical classes | 6.12; P.40; P.45; P.43; P.41 |
| | "Remove constrictive items from bitten limb" | 4 | Paramedical classes | P.44; P.45; P.46; P.41 |
| | "Remove tight clothing from body of the victim" | 1 | Paramedical classes | P.44 |
| | "Examine the airway. Provide **artificial respiration** and apply chest compression for cardio-pulmonary resuscitation if necessary" | 6 | 6, 7 | 6.12; 7.19; 7.20; 7.15; 7.16; 7.18 |
| | "Tell doctor about systemic symptoms evolved on the way to hospital", "monitor the symptom development" | 4 | Paramedical classes | P.44; P.45; P.46; P.41 |
| | "manage shock if present" | 1 | Paramedical classes | P.41 |
| | "Do not apply tight bandage (**tourniquet**)" | 9 | 7, B.Ed., Paramedical classes | 7.15; B.39; P.40; P.44; P.45; P.46; P.43; P.41; P.42 |
| | "Do not **cut** wound to let the venom come out with bleeding" | 8 | 7, B.Ed., Paramedical classes | 7.15; B.39; P.44; P.45; P.46; P.43; P.41; P.42 |
| | "Do not **suck** blood from the wound to keep out venom" | 9 | 7, B.Ed., Paramedical classes | 7.15; B.39; P.40; P.44; P.45; P.46; P.43; P.41; P.42 |
| | "Do not allow **eating** and **drinking** or other stimulant such as alcohol, chillies, etc.", "do not give pain killing medicine unless doctor suggest for this medicine" | 6 | 6, 7, Paramedical classes | 6.12; 7.17; P.45; P.46; P.41; P.42 |
| | "Do not interfere with bite site" | 2 | Paramedical classes | P.40; P.44 |
| | "Do not apply chemicals or electricity on bite site" | 3 | Paramedical classes | P.40; P.44; P.42 |
| | "Do not apply cooling agent or cold compress to the wound" | 5 | Paramedical classes | P.44; P.45; P.46; P.43; P.42 |
| | "Do not keep bitten part above the position of heart" | 3 | Paramedical classes | P.45; P.46; P.41 |
| | **RAFA**: "immediately carry or transport victim to the nearest healthcare institution supplied with anti-snake venom for medication of envenoming" | 4 | 7, Paramedical classes | 7.20; 7.19; P.41; P.42 |
| *NRA* | "Keep bitten part **below the position of heart**" | 6 | 6, Paramedical classes | 6.11; 7.17; P.44; P.45; P.46; P.41 |
| | "Apply tight bandage (tourniquet) above the bite" | 6 | 6, 7 | 6.12; 7.19; 7.20; 7.17; 7.16; 7.18 |
| | "Make incisions through the skin by sterilized knife or blade and let the venom come out" | 2 | 7 | 7.16; 7.18 |
| | "Apply suction cup, if possible; otherwise use mouth to suck blood continuously." | 1 | 7 | 7.16 |

*(Continued)*

**Table 1.** (Continued)

| Themes | Quotes/statements indicating advice for first aid of snakebite | Books (N = 16) | Classes using books | Secondary codes of books (Please, see the S2 Table for the further details) |
|---|---|---|---|---|
| | "Apply ice packs or cold water to the wound" | 2 | 6, B.Ed. | 6.12; B.39 |
| | "Give warm tea or coffee to the victim" | 1 | B.Ed. | B.39 |
| | "Bring dead snake to hospital along with the victim if it is easy to kill snake" | 2 | Paramedical classes | P.45; P.46 |
| *PAA* | "Carry the victims right away to a doctor or a hospital/health center/health post" | 12 | 6, 7, B.Ed., Paramedical classes | 6.11; 6.12; 7.15–7.18; B.39; P.40; P.44–P.46; P.43 |

(S1 Table). These were the national textbooks for the entire country although we selected them from educational institutions located in Nepal's two districts. We excluded "Maths", "Physics", and other reference books.

### Data coding and their clustering and interpretations

We evaluated the advice for first aid for snakebites mentioned in those 46 textbooks by developing themes which we coded and quantified according to the previous similar study [36]. For this, we manually created clean, readable photocopies of the transcripts of "Safety and First Aid" unit of those books, assigned the book codes (S2 Table) over the photocopied transcripts, read each transcript several times to get a general understanding of the mentioned first aid, selected quotes/statements that were relevant to the first aid of snakebite, and assigned particular codes (RA, RAFA, NRA, PAA, and error) to $i^{th}$, $ii^{th}$,... $n^{th}$ quotes (i.e., advice for first aid of snakebite). We categorized those advice and clustered them into three themes (Table 1): recommendable advice (code: RA), non-recommendable advice (NRA), and partially accurate advice (PAA) based on the similarities or relatedness of standard advice for snakebite first aid available in published guidelines of the WHO [31]. We verified and graded those themes in reference to these widely used guidelines.

We used code RA for recommended advice such as using pressure-immobilization bandaging (PIB) or local compression-pad immobilization (LCPI), to go to an appropriate facility provided with anti-snake venom [subcode of RA: RAFA (Yes = mentioned, No = not mentioned)], NRA for non-recommended advice, and PAA for partially accurate advice (i.e., advice to carry victims to a doctor or to healthcare facilities without defining about the need of antivenom would be supplied therein). We counted advice appropriate to RA, RAFA, NRA, and/or PAA for quantitative analyses of frequencies for respective categories (Table 2 and Table 3).

We expressed 100% RA if all advice for first aid were compatible (or verifiable) with the WHO guidelines [31]. Similarly, we expressed 100% NRA if all advice were non-standard. The RAFA was evaluated in terms of "yes = mentioned" and "no = not mentioned". The total erroneous advice (i.e., overall errors of respective textbooks) was the sum of percentages for NRA and PAA.

### Hypotheses and data analyses

We hypothesized that the proportions of RA was greater and NRA, PAA, and total erroneous advice were less in books that we evaluated than those evaluated previously [36]. We expected significant difference in proportions for RA, NRA, PAA, and total errors contained in the books evaluated in the past and at present.

We analyzed snakebite first aid in textbook by quantifying the themes in terms of ranges, medians, and inter-quartile ranges (IQR) of scaled first aid (in percentage). We compared the median proportions (herein, percentages) of overall RA, NRA, PAA, and total errors and counts of books containing RA, NRA, and PAA known in this study with the previous similar study performed in Nepal [36] by using the unpaired two-samples Wilcoxon test with continuity correction and two-proportions Z-test for equality of proportions with continuity correction at 5% significance level, respectively, to find

**Table 2. Scaled commission of inappropriate first aid measures in textbooks containing "Safety and First Aid" unit.**

| Secondary codes | Class | Textbook´s name/ medium (details about books and codes are in S2 Table ) | Year | Author/s | Page ranges devoted to snakebite health education | First aid [NA: not mentioned; i, ii,..: 1st, 2nd,... advice for first aid] | | | | | | |
|---|---|---|---|---|---|---|---|---|---|---|---|---|
| | | | | | | RA (orange and bold text is washing bite site) | RA% | RAFA | Errors (red and bold text advice is position of bitten part below the level of heart.) | | | |
| | | | | | | | | | NRA | NRA% | PAA | PAA% | Total (%) |
| 7.18 | 7 | HPE&CA */ Eng. | 2022 (2023, 2024) | TNB | 111–112 | i, **iv**, vi | 43 | No | ii & iii (both tour-niquet), v | 43 | **vii** | 14 | 57 |
| 6.11 | 6 | HPE&CA */ Eng. | 2021 (2023) | BDK, RM | 100 | **i**, iii | 50 | No | **iv** | 25 | **ii** | 25 | 50 |
| 7.17 | 7 | HPECA/ Eng. | 2022 (2023) | BDK, RM | 98 | i (1st section), ii, iv | 50 | No | **i** (2nd section), iii | 30 | **v** | 20 | 50 |
| 7.16 | 7 | HPE&CA */ Eng. | 2022 (2023) | Aryal et al. | 80–81 | i, ii, v, vi, ***vii (1st section)*** | 64 | No | iii, iv | 29 | **vii (2nd section)** | 7 | 36 |
| 6.12 | 6 | HPE&CA */ Eng. | 2024 | TNB | 69, 72–73 | i, ii, **iv**, vi, vii, ix, x | 70 | No | iii, v | 20 | **viii** | 10 | 30 |
| B.39 | B. Ed. 4th year | School Health Program and Community Health Survey/ Nepali | 2019 | CBB, BPW | 87–88 | i–iii, vi, **vii**, **viii**, ***ix*** (1st section), x | 75 | No | iv, v | 20 | **ix (2nd section)** | 5 | 25 |
| 7.20 | 7 | HP&CA **/ Eng. | 2022 | CDC | 56–57 | i, **ii**, iv, **v** | 80 | Yes | iii | 20 | 0 | 0 | 20 |
| 7.19 | 7 | HP&CA **/ Nep. | 2023 | CDC | 53–54 | i, **ii**, iv, **v** | 80 | Yes | iii | 20 | 0 | 0 | 20 |
| P.45 | Para-medical | TFN ‡‡/ Eng. | 2021 | SP, RD | 524–527 | i, ii (1st section), iii, iv, **v**, viii–xvi | 84 | No | ii (2nd section), vii | 9.4 | **vi** | 6.3 | 16 |
| P.46 | Para-medical | TFN ‡‡‡/ Eng. | 2019 | SP | 501–505 | i, ii (1st section), iii–**vi**, ix–xvii | 85 | No | ii (2nd section), viii | 9 | **vii** | 6 | 15 |
| P.40 | Para-medical | Essential Textbook of Basic Medicine-I/ Eng. | 2022 | KG, AB | 371–372 | i–**iii**, v-viii | 88 | No | 0 | 0 | **iv** | 13 | 13 |
| 7.15 | 7 | HPE&CA*/ Eng. | 2024 | AKB | 102–104 | i, **ii**, **iii**, iv, vi–viii | 88 | No | 0 | 0 | **v** | 13 | 13 |
| P.44 | Para-medical | A Textbook of Fundamental of Nursing ‡‡/ Eng. | 2021 | GNM, DPS | 417–419 | i,**ii**, iv–xi, **xii**, xii–xiv, **xvi** | 88 | No | **xv** | 6.3 | **iii** | 6.3 | 13 |
| P.43 | Para-medical | Essential Textbook of Basic Medical Procedure and First Aid (ETBMP & FA)/ Eng. | 2019 | IG | 224–229 | i-**iv**, vi, **vii**, viii-x | 90 | No | 0 | 0 | **v** | 10 | 10 |
| P.41 | Para-medical | A Textbook of First Aid and Basic Medical Procedure/ Eng. | 2020 | SS, SP | 200–201 | i-ii, iii (1st section), iv–**viii**, ix–xix | 97 | Yes | iii (2nd section) | 3 | 0 | 0 | 3 |
| P.42 | Para-medical | A Textbook of Basic Medical Procedure & First Aid ‡/ Eng. | 2018 | TP | 129–133 | i–viii, **ix**, x | 100 | Yes | 0 | 0 | 0 | 0 | 0 |
| Median (Inter-quartile range) | | | 2022 (2020–2022) | | | | 82 (69–88) | | | 15 (2–21) | | 7 (4–13) | 18 (13–32) |

**Abbreviations and symbols used: Class**: ***B.Ed.*** = Bachelor of Education, **Ist yr.** = First year, **IInd yr**. = Second year, ***Paramedical*** = Aayurvedic Health Assistant, Aayurvedic Auxiliary Health Worker, Bachelor in Pharmacy, Bachelor in Public Health, B.Sc. Nursing, Bachelor of Nursing, BAMS: Bachelor of Aayurvedic Medical Science, BDS = Bachelor of Dental Surgery, Community Health Worker, Community Medicine Assistant, Diploma in Pharmacy,

*(Continued)*

**Table 2.** (Continued)

Health Assistant, Proficiency Certificate Level (**PCL**) of Nursing, PCL of Pharmacy; **Year:** Latest edition year (reprinted years are in parenthesis); **Year of latest edition of textbooks**: Publication year mentioned in Nepali year [Bikram Sambat (BS)] system is generalized as: 2075 BS = 2019 AD, 2076 BS = 2020 AD, 2077 BS = 2021 AD, 2078 BS = 2022 AD, 2078/2079 BS OR 2079 BS = 2023 AD, 2080 BS = 2024 AD; **First aid:** basic and immediate actions that anyone can perform to stabilize a victim before professional help arrives; **RA** = recommended advice [i.e., i) advice to use **PIB** or **LCPI** are bold, ii) advice to emergency transport of snakebite patients in ambulance or appropriate vehicle in the column "**RA**" is bold and italicized, and iii) **RAFA** (i.e., recommendation to go to an appropriate healthcare facility provided with anti-snake venom is measured in terms of Yes (mentioned) and No (not mentioned) information); **NRA** = non-recommended advice; **PAA** = partially accurate advice (advice to carry victims to a doctor or to undefined health facilities is bold); **Page ranges**: pages containing advice for first aid. **Eng.** = English; **Nep.** = Nepali; **Author/s:** Please, see S2 Table for full forms of abbreviated authors.

**Table 3.** Comparison of percentages (proportions) of advice for first aid of snakebite and total errors contained in books evaluated in the past[1] and during this study.

| Advice for first aid mentioned in the selected books | | Books evaluated [N: number of books] for first aid of snakebites | | | | Comparing the advice in the past and at present books | |
|---|---|---|---|---|---|---|---|
| **A] Median and inter-quartile range (IQR) of percentages for RA, NRS, PAA, and total errors in a set of advice for first aid mentioned in all evaluated books** | **Cohen's d (i.e., effect size)‡** | **in the past[1] (N = 31)]** | | **at present (N = 16)** | | **Unpaired two-samples Wilcoxon test*** | |
| | | Median percent | IQR percent | Median percent | IQR percent | p-values (2 tailed) | p-values (1 tailed) |
| **a. RA** (recommended advice) | 2.378 | 29 | 25–38 | 82 | 69–88 | <0.001 | <0.001 (right) |
| **b. NRA** (non-recommended advice) | 2.463 | 57 | 50–65 | 15 | 2–21 | <0.001 | <0.001 (left) |
| **c. PAA** (partially accurate advice) | 0.554 | 14 | 11–16 | 7 | 4–13 | 0.033 | 0.017 (left) |
| **d. Total errors** | 2.371 | 71 | 63–75 | 18 | 13–32 | <0.001 | <0.001 (left) |
| **B] Percentages for RA, RAFA, NRA, and PAA in a set of advice for first aid mentioned in particular evaluated books** | **The effect size (for book counts compared in %)** | **Books' count (N = 31)** | **Percent** | **Books' count (N = 16)** | **Percent** | **Two-proportions Z-test*** | |
| **B.a. RA** | | | | | | p-values (2 tailed) | p-values (1 tailed) |
| i. Applying PIB or LCPI | 0.156 | 2 | 6 | 11 | 69 | <0.001 | <0.001 (right) |
| ii. Emergency transport in ambulance or appropriate vehicle | | 17 | 55 | 2 | 13 | 0.013 | 0.006 (left) |
| iii. Recommendation to go to an appropriate healthcare facility provided with anti-snake venom (i.e., RAFA) | | | | | | | |
| RAFA: Yes | | 5 | 16 | 4 | 25 | 0.733 | 0.367 (right) |
| RAFA: No | | 26 | 84 | 12 | 75 | 0.733 | 0.367 (left) |
| **B.b. NRA** | | | | | | | |
| 1. Incision | 1.311 | 27 | 87 | 2 | 13 | <0.001 | <0.001 (left) |
| 2. Wound (bite site) sucking | | 12 | 39 | 1 | 6 | 0.044 | 0.022 (left) |
| 3. Ligature/tourniquet application | | 24 | 77 | 6 | 38 | 0.017 | 0.009 (left) |
| 4. Providing liquid/water or stimulants to the victim | | 17 | 55 | 1 | 6 | 0.003 | 0.002 (left) |
| 5. Application of an icepack or cold water | | 7 | 23 | 2 | 13 | 0.659 | 0.330 (left) |
| 6. Keeping the bitten part below the level of heart | | – | – | 6 | 38 | – | – |
| 7. Excluding necessary information about the availability of antivenom (PAA†) | | 25 | 81 | 12 | 75 | 0.943 | 0.471 (left) |

[1]cited in: Pandey and Khanal 2013; *Comparing the proportions with Yates' continuity correction to determine if there is a significant difference of associated proportions in the past and at present textbooks. †advised partially correct practices, which is suggestion for carrying snakebite victims to a doctor or to health facilities without disclosing about the specialty of the healthcare institutions or antivenom availability. ‡expressed values in standard deviation unit.

any changes in median proportions of associated advice and errors in the past and present books and frequencies of those books containing associated advice and errors (Table 3). Further, we used Cohen's d to measure the size of the difference between respective proportions of two groups of books. We interpreted the usefulness of these textbooks based on recommended advice without errors and the measure of effect size. We interpreted a value of Cohen's d as 0.2 to 0.4 or less for a small effect size, 0.5 to 0.7 for a medium effect size, and for 0.8 or more for a large effect size. As the effect size indicates the practical significance of a research outcome, we interpreted the large effect size as a research finding having practical significance, while a small effect size as limited practical applications.

All statistical analyses were performed using R-Statistical Programming (R version 4.4.0). We mentioned p-values below 0.001 as <(less than) 0.001.

## Results

### Proportions of books with/without first aid of snakebites

We evaluated 46 textbooks that were edited during 2018 and 2024 (the median year of latest edition was 2022; Fig 1). The 38 out of 46 books were used by students enrolled for school curricula (i.e., by 4–12 graders), seven books by paramedical students enrolled for the CTEVT and university curricula, and one book used by students enrolled for Tribhuvan University curricula (S1–S2 Tables).

A total of 11 (24%) textbooks [used by 11th graders (Health and Physical Education, i.e., HPE) and 11th and 12th graders (biological books)] contained no "Safety and First Aid" unit whereas 35 (76%) textbooks mentioned the "Safety and First Aid" unit (S2 Table). Interestingly, 19 out of these 35 (54%) textbooks [four different authored "Health, Physical (Education) and Creative Arts (i.e., HP (E) & CA)" books published in English and Nepali language edition, each used by 4th–6th graders, respectively, six different HP (E) & CA used by 8th grader, and one HPE used by 12th grader] did not advise for the measures of first aid of snakebites (S2 Table). Similarly, the "Health and Population" book used by 5th, 6th, 8th, and 11th graders, the "Health, Population, and Environment, hereafter, HPEn" by 9th graders, the "Introductory HPEn" and the "Fundamental Health and Physical Education" by 11th graders, and the "Foundation of Health" and the "Basic Health Science" by B.Ed. first and second years' students, respectively, contained first aid measures previously [36]. While evaluating similar books, we found recently edited textbooks used by 4th, 5th, 6th (except two English edition books), 8th, and 12th graders without first aid measures of snakebite despite containing "Safety and First Aid" unit (S2 Table).

The rest of 16 (46%) recently evaluated textbooks used in grade 6th, 7th, B.Ed. 4th year, and paramedical sciences prescribed a median of 18% erroneous first aid, i.e., the mentioned first aid advocated in these textbooks excluded appropriate measures recommended in published guidelines, and recommended inappropriate first aid. Among the currently used textbooks in Nepal, the "Safety and First Aid" unit was included starting the textbooks prescribed for the 4th graders (S2 Table). But, the advice for snakebite first aid was included in a couple of English edition textbooks prescribed for 6th graders (Table 2, Fig 1). Although HPE used previously by 11th graders contained first aid measures [36] and our key informants supposed to contain first aid in 11th and 12th graders' biological books (S2 Table), these textbooks did not include "Safety and First Aid". The snakebite first aid aimed to B.Ed. first and second years' students [36] was moved to the books used by B.Ed. 4th year students (Table 2).

### Precision of the first aid of snakebites mentioned in the textbooks

Only one textbook was the most useful without errors and containing adequate standard advice for first aid after snakebite. The most important and commonest advice was to reassure victims making them calm with emotional support (13 out of 16 books contained this advice). Only 11 out of 16 (69%) textbooks recommended applying PIB or LCPI. Emergency transport of snakebite patients in ambulance or appropriate vehicle was recommended in 2/16 (13%) of textbooks. Four out of 16 textbooks (25%) suggested going to an appropriate healthcare facility provided with anti-snake venom for snakebite treatment (Table 2).

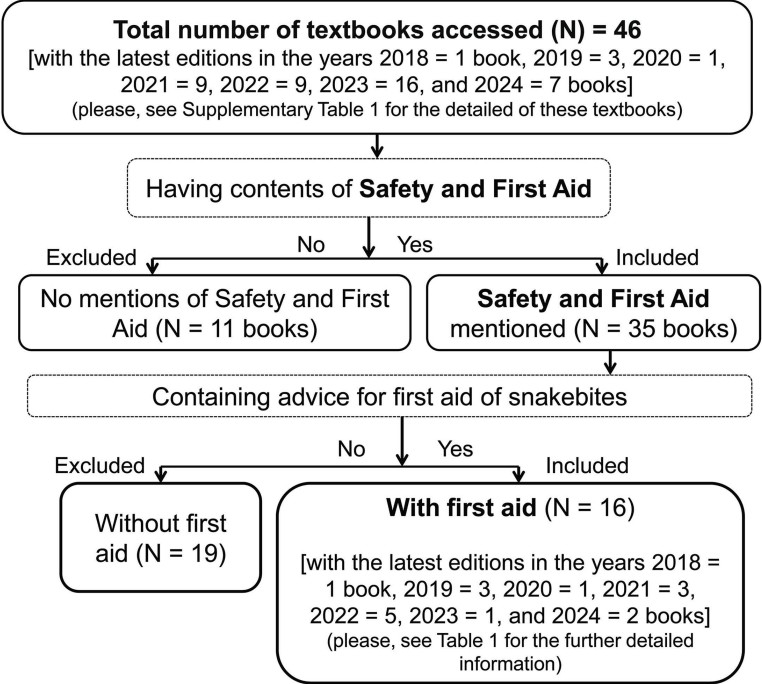

**Fig 1. Flow-chart showing selection of textbooks in Nepalese educational institutions and containing measures for first aid of snakebites.**

### Inappropriate first aid of snakebites

Except a book with zero errors, a total of 15 books still contain certain errors (Table 2). Useless or unverified and/or harmful advice included incision [quoted as: "Make incisions through the skin by sterilized knife or blade and let the venom come out"] in two out of 16 textbooks (13%), wound sucking by mouth and/or suction pump ["Apply suction cup, if possible; otherwise use mouth to suck blood continuously."] in one out of 16 (6%), ligature/tourniquet application ["Apply tight bandage (tourniquet) above the bite"] in six out of 16 (38%), providing liquid/water or tea, milk, coffee to the victim ["Give warm tea or coffee to the victim"] in one out of 16 (6%), application of an icepack or cold water ["Apply ice packs or cold water to the wound"] in two out of 16 (13%), and keeping the bitten part below the level of heart ["Keep bitten part below the position of heart"] in six out of 16 (38%). Of the 16 textbooks, 12 textbooks (75%) excluded necessary information about the availability of antivenom nearest to the activity areas of people vulnerable to snake-bites and advised people to carry victims to a doctor or to health facilities without specifying the specialty for snakebite care (Tables 1–2).

### Comparison of advice for first aid

The median percentage of RA increased and errors (i.e., NRA, PAA, and total errors) decreased in books evaluated during this study significantly (p < 0.001–0.017, Table 3, Figs 2–3). The incision [p < 0.001], wound sucking by mouth and/or suction pump [p = 0.022], ligature/tourniquet application [p = 0.009], and providing liquid/water to the victim [p = 0.002] advised in previous set books were noticeably reduced the set of books evaluated during this study (Table 3.B). The average RA in previously evaluated books was 2.378 standard deviations below the average RA in books evaluated during this study, whereas the average errors in previously evaluated books was 2.371 standard deviations above the average errors in books evaluated during this study (Table 3.A). All suggested improvement in the recently used textbooks in Nepal.

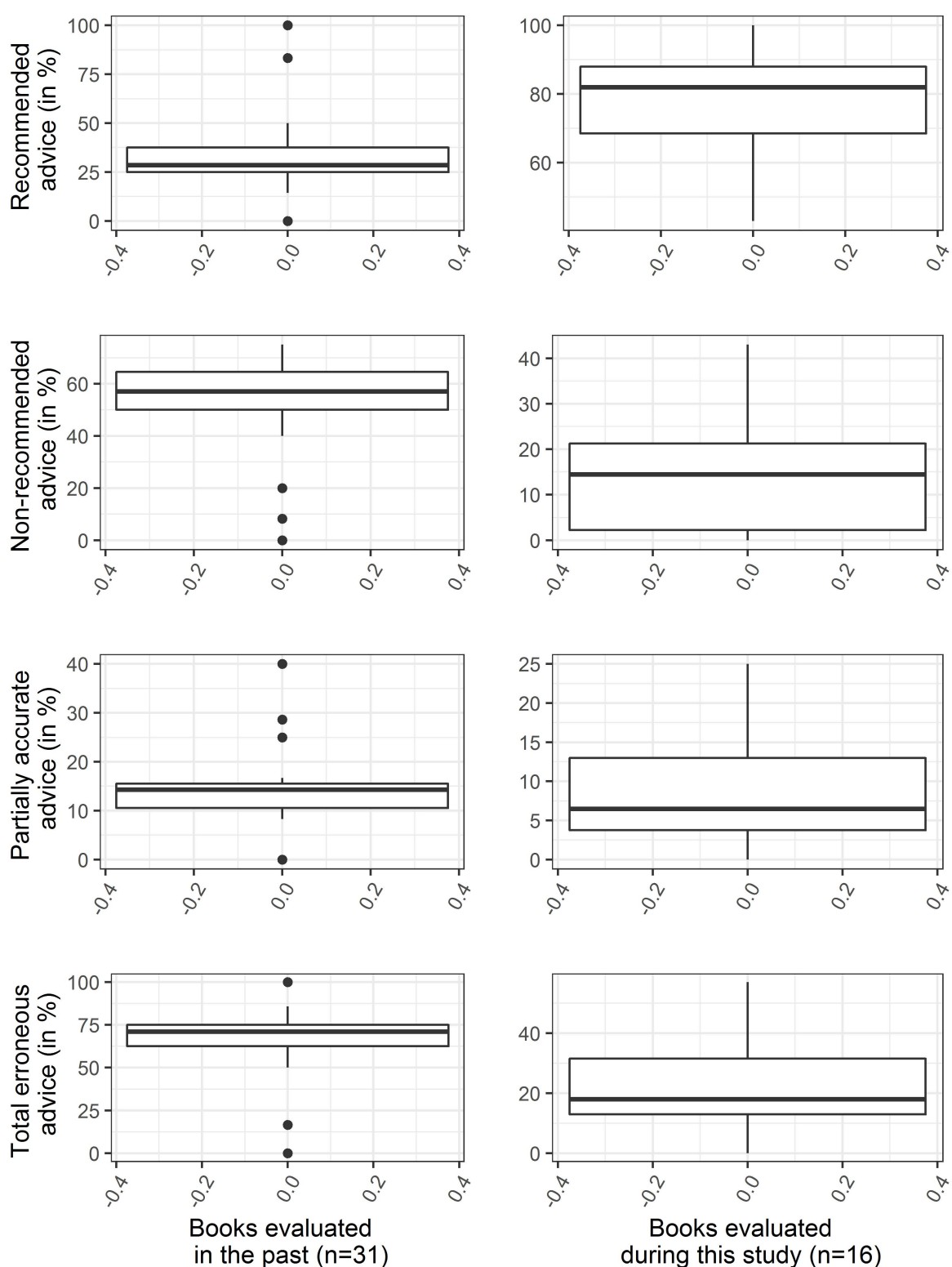

**Fig 2. Box plots showing comparison of percentages of recommended, non-recommended, and partially accurate advice as well as total erroneous advice mentioned in textbooks evaluated in the past and at present.** The medians of advice for standard first aid increased and for substandard or harmful first aid decreased.

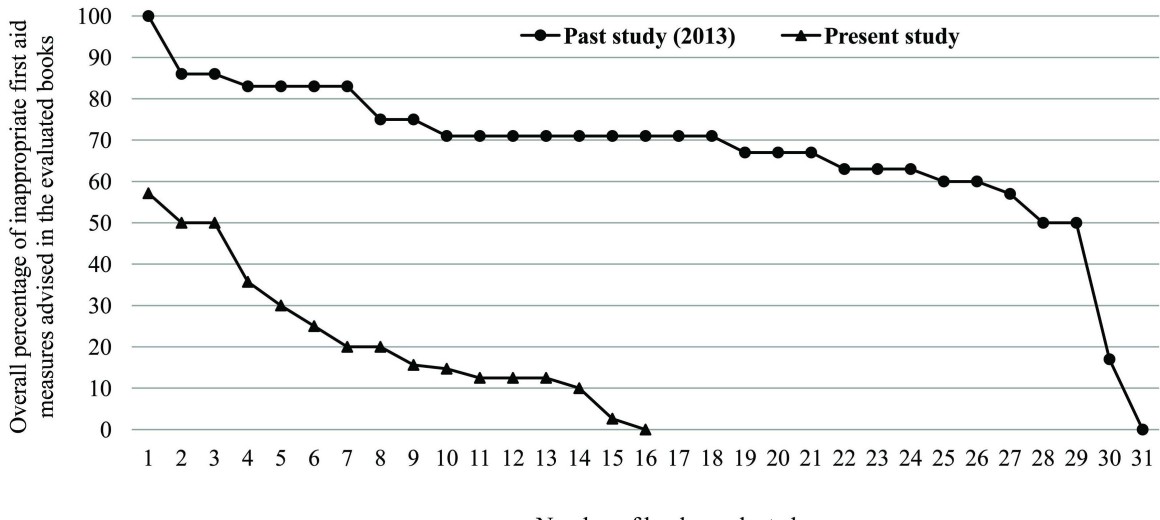

**Fig 3. Overall percentage of inappropriate first aid measures recommended in the textbooks evaluated (i.e., total errors) in the past and present study.**

However, two-proportions Z-test showed no evidence to support any difference of proportions for books advising to apply an icepack or cold water on bite site (p = 0.659) and for exclusion of information on need of knowing availability of antivenom nearest to the residential areas (p = 0.943) in books evaluated in the past and during this study (Table 3.B). Number of textbooks advising partially correct practices known in the past and present study was not statistically different (p = 0.943, Table 3). Further, average of proportions of 2013 books containing NRAs was 1.311 standard deviation above the average for the proportions of 2024 books containing NRAs (Table 3.B). This effect size suggested the persistent teaching of harmful snakebite first aid in Nepal.

While comparing components of RA, we found significant increase in number of books advising snakebite victims to apply PIB or LCPI, emergency transport of snakebite victim in ambulance or appropriate vehicle, and to go to an appropriate healthcare facility provided with anti-snake venom (p < 0.001–0.006, Table 3, Fig 2). But, there existed noticeable differences between numbers of books in the past and recent evaluation (p = 0.013, Tables 1, 3), with highlights for the need of emergency transport of snakebite victims using ambulance or comfortable vehicle in small number of books being currently used in Nepal. Further, the overall effect size was small (0.156 standard deviation) while comparing proportions of books containing RAs (Table 3.B). This suggested the actual difference between the averages for RAs of both set of books to be trivial.

### First aid measures appropriate for a particular level of formal education

There was no distinction between what first aid measures included in a textbook for 4th/6th/7th grade students compared to the information included in a textbook for a bachelor degree and paramedical classes. The advice would be according to the level of mental growth, understanding, and grades.

### Discussion

Large effect size (Table 3) of recommended and non-recommended "first aid measures" indicated improvement in currently evaluated textbooks. This improvement could be the influence of the past publication [36] that recommended to include appropriate first aid in books used by Nepalese students, similar advice in newspapers [19], and interviews

of DPP (first author of this article) published in newspapers, radios, and televisions and invited trainings in which DPP shared the need of improvement in textbooks used in Nepal. However, there was persistent teaching of harmful or non-indicated interventions for first aid of snakebites [13–32% inter-quartile range of erroneous first aid, Table 3]. The first aid advocated in those textbooks excluded appropriate measures recommended in published guidelines [7,31] and recommended inappropriate first aid (Tables 1–2) although some of non-standard first aid are known to be harmful [12,15]. Incision and sucking of the snakebite wound (by using mouth or any other suction device) to let the injected venom out are worthless [30,37], cause the local tissue damage/injuries [38,39], and keep victims at the risk of heavy bleeding. Similarly, the use of ligature/tourniquet is harmful resulting in amputation of ligatured limb [40]. Although use of icepack or cold water causes more harms than benefits [41], it is still suggested to use in the recently used books. Similarly, the exclusion of necessary information about the availability of antivenom in snakebite risk areas (Tables 1–2) can mislead the snakebite victims to a doctor or to health facilities without specialty of the healthcare institutions or antivenom availability. This omission was the next barrier to snakebite management because the recommendation of going hospital without confirming the availability of antivenom and without encouraging to use the most comfortable and quickest mode of transport [42] was incomplete and might cause delay in definitive care of envenomated patients because all healthcare institutions have no antivenom or sufficient facilities or trained healthcare professionals for managing envenomed cases [20,25]. Emergency transport of snakebite patients with severe venom effects in ambulance or appropriate four-wheel vehicle is more beneficial than transporting in motorcycle because it is easier to immobilize the whole body of the snakebite victim, with the recovery position to protect airway of severely affected patient, to facilitate expelling saliva and vomit [31]. Otherwise, entry of saliva and contents of vomit into the respiratory track increases risk of choking and pneumonia.

The passive immunotherapy with safe and effective antivenom is the only effective treatment of snakebite envenoming [43,44]. This prevents severity of venom effects (e.g., paralysis, acute kidney injury, and death) and increases optimum outcome if antivenom is administered approximately 1–2 h post-snakebite [43,45]. Among the 61% (n = 44) envenomed cases who presented to STCs after 4 hours of snakebite, 30% (n = 13) patients died or disabled in Nigeria [15]. Therefore, since time interval is critical to improve patient outcomes, the public awareness of the existence of antivenom and hospitals which stocked it in case of emergency is highly essential. Considering this need, authors should edit their textbooks advising book-readers to receive treatment with antivenoms within one hour of snakebite [46] and health policy makers to ensure the supply of antivenom to healthcare institution within an accessible distance from the origin of snakebites to minimize the deaths and disabilities due to snakebites. Overall, the concurrence of our findings with the published data [36] reflected the persistent teaching of erroneous first aid in Nepal although erroneous advice for first aid was decreased in the currently used books (Fig 3, Table 3). This persistency in teaching of erroneous first aid may confuse teachers and students (or community people) to accept the first aid measures recommended by the WHO [31].

Inadequate monitoring of contents in currently used textbooks by the CDC, the CTEVT, and the universities as well as likely dependence of books' authors on recommendations of inappropriate measures available in websites developed by non-experts in snake and snakebites [e.g., 26 (54.1%) of 48 websites provided inaccurate first aid of snakebites [47]] might be possible reasons to authors including incorrect information or inadequate information in their textbooks being used in Nepal. This persistency of harmful or non-indicated first aid of snakebites in textbooks indicated the neglect of authors and/ editors of the respective books and poor monitoring of quality of curriculum texts by formal education sectors. These neglects are barriers to effective snakebite management because these poor and incomplete textbooks and non-updated websites promote non-recommended measures and cause continuation of using harmful or useless first aid in Nepal and elsewhere. Therefore, except a book with zero errors, a total of 15 books need re-edition although the advice of harmful, useless or non-standard measures in textbooks evaluated during this study were greatly reduced (p < 0.001–0.022, Table 3.A,B, Figs 2–3). In Nepal, the common sources

of lacking knowledge on the standard first aid were school books (that provided flawed or outdated knowledge on snakebites [36,48]) used during their student-tenure. In a cross-sectional study carried in Nepal, it is reported that 77.8% medical students erroneously believed applying tourniquet [48]. Similar ignorance of medical and non-medical students and others inhabiting Nepal [23,24,49] and elsewhere [13,21,50–61] in snakebite prone communities would also be the results of inaccurate or inadequate information on snakebite management mentioned in the inadequately edited textbooks or reference books which they used during their student-tenure [36]. The lack of educational materials required to use first aid options after snakebite elsewhere indicated that the problem of teaching inadequate information to students is not unique to Nepal. Additionally, the inclusion of these harmful interventions in the teaching materials may propagate the extremely risky practice at community level, too. In these contexts, the re-edition of those recently used 15 textbooks should be with reference to what ought to be/is appropriate for a particular level of formal education about first aid and prevention of snakebites because the advice should be compatible with the need and mental growth of students and be supportive to prevention and control of snakebites effectively. For example, information about antivenom, need of supply and availability of antivenom at a healthcare centre/hospital within one-hour-motor-distance from areas at risk of snakebite envenoming, how to transfer a patient to the referral healthcare facility supplied with antivenom, and application of first aid requiring special skills and materials (e.g., PIB, LCPI, Box 1) was not appropriate for the basic level students. Rather, these measures requiring intensive teaching/training and specific materials to develop special skills were appropriate advice for secondary and university level and paramedical students. These measures requiring intensive teaching to develop special skills using specific materials (for example: PIB, LCPI) and those requiring no intensive teaching are presented in Box 1. The textbooks prescribed for non-medical (basic, secondary, and university level) and paramedical education can advise the measures listed in the Box 1 according to the need and mental growth of students.

Since the consequence of adopting harmful or useless interventions is expensive [15,16] and wasting time increases the risk of death or complications [e.g., infections on bite site, amputation of the affected organ] [15,19], knowing appropriate first aid is imperative to save lives and reduce complications by timely managing of SBE. We could not access published research articles to know what is the situation of the first aid advocated in textbooks (excluding appropriate measures recommended in widely used guidelines and suggesting inappropriate first aid although some of non-standard measures are harmful) used in other countries. However, this study can be a basis for similar evaluation and improvement of textbooks used in other snakebite prone nations worldwide.

The recent removal of first aid unit or snakebite first aid subsection from the first aid unit does not align with the WHO's goal of halving snakebite burden by 2030 [3,62]. Since children are highly vulnerable group of snakebites in the context of Nepal and elsewhere [63], first aid practices should be included in all level of textbooks used for school education. Students should be enriched with reasons for recommended and non-recommended practices gradually while increasing the level of grades. In the higher secondary level education, they should have developed the ability to share about first aid with their family members and neighbors and apply it when it is needed. Similar empowerment of healthcare workers by formal education [62] is the next key to support the WHO's objective of halving snakebite deaths and disabilities by 2030. In this way, curriculum texts of first aid of snakebites should be improved to meet with the public needs. Hence, the measures for first aid of snakebites should be retained in grade four to university classes according to increased ability of students in the upper classes to encourage vulnerable people for rapid actions on getting pre-hospital and in-hospital care of snakebites [45]. This study encourages textbook writers, teachers, students, and health and education policy makers to follow up the standard first aid of snakebite (Box 1) although several international to local and formal to informal educational interventions in the past reduced the mentions of inappropriate first aid of snakebites and increased the advice for the standard first aid of snakebites over time (Figs 2–3, Table 3). Further improvements in texts of first aid (Table 1 and Table 2, Box 1) contained in textbooks are imperative to nullify the errors and increases the level of knowledge of students and teachers

on first aid, which in turn influences community health. It is because this study finding is potentially impactful to more than 7,106,788 students nationwide primarily. Next, the school and university students and teachers are significant demographic components influencing community health by disseminating approved practices and increasing the awareness for snakebite management. Since the evaluated books during this study are national textbooks being used nationwide, this study finding is important for the authors, editors, and readers of associated books and also for government officials. Since community factors (i.e., inappropriate first aid methods and delayed arrival at health facilities) and/or health system factors (i.e., antivenom accessibility and availability of competent healthcare providers) influence on snakebite outcomes [64], there is an urgent need of revising those textbooks (S2 Table) and improving formal education of snakebite, too. Our list of educational materials with specific advice and errors can be useful to edit the corresponding units of the associated textbooks (S2 Table, Table 1, Box 1). Since the WHO guidelines for the management of snakebites in Asia suggests including snakebite prevention and management measures in the curriculum of medical and nursing schools [31] and highlights the need of health education to children and young people, particularly 10–19 year old age group, due to high incidence of snakebite at this age interval, the measures suggested in the widely accepted and used WHO guidelines should also be followed up by associated authors while re-editing those textbooks, writing books containing suggestions for prevention and management of snakebite, and increasing the awareness of effective first aid (with the notes for special attention while using PIB and LCPI) among students and locals inhabiting snakebite prone zones in Nepal. The improved textbooks using regional [31] and country-specific snakebite management guidelines [7,8] should emphasize the timely first aid and pre-hospital management of venomous snakebites by potential bystanders [i.e., students, teachers, and paramedical personnel (who are mostly deployed in semi-urban and rural healthcare institutions) having sufficient knowledge on how to give first aid] to contribute in decreasing incidence of snakebite, improving survival rates, and minimizing long-term complications after a snakebite or reducing complications due to adoption of useless or potentially harmful first aid of SBE in disadvantaged communities in the tropics and the sub-tropics of Nepal and other countries [18] where formal education of snakebite is inadequate.

The recent research demonstrates a significant gap in public knowledge about appropriate first aid measures among diverse communities and occupations. Similar to aforementioned neglect of Nepalese education and health sectors, the widely used tourniquet is one of influencing barriers in other nations in Asia and Africa [12,40,51,60] despite public/community education on not to apply tourniquets. In the eastern India, three-quarters of all snakebite victims lacked knowledge of the required first aid measures following snakebite [65]. Less than 5% (n = 33) community health workers and community members had adequate knowledge of first aid measures in rural Malawi [59], nursing and medical students of Palestine had mean first aid knowledge score as 6.6/15 (44%) and 8.3/15 (55.3%), respectively [53,54], only 39% community people knew about the correct methods of first aid in Myanmar [51], and none of the farmers from North Central Province of Sri Lanka had knowledge of pressure-bandaging and immobilizing [66] whereas nearly 50% Sri Lankan parents immobilized the affected limb of their children after snakebite [60]. A study carried in Nigeria reported that the application of tourniquet increased the median costs of hospitalization and risks of wound infections and negatively affecting the outcome of treatments at hospital [15]. Therefore, it is an urgent need to enact policies to provide standard first aid in textbooks being used in schools and universities of Nepal and elsewhere and educate people in snakebite prone communities involving the recently revised textbooks based on reliable scientific publications because an appropriate education on first aid for venomous snakebite improves outcomes of in-hospital treatment of snakebites [15,31] and increases the feeling of social security among people inhabiting the snakebite prone regions. All these underscore the need for strengthening first aid responses in schools and universities, and technical institutions dedicated to paramedical education worldwide. It is because school and university students can share measures against snakebite and first aid skills to their peers, parents, relatives, and neighbors vulnerable to snakebites.

Although there are evidence of ability to learn basic first aid and prevention measures by the kindergarten children aged 4–5 years [67] and snakebite is a serious, time critical health issue affecting mainly children [68,69] due to their susceptibility to severe envenoming (resulting in respiratory failure, renal failure, and even death) because of their smaller body size [69], and careless behavior (such as playing in areas with snake-hiding places) of children, the learning of snakebite first aid in schools of Nepal mainly confined to 7th grader (expect two English edition "Health, Physical, and Creative Arts" targeted to 6th grader contained snakebite first aid measures; Table 2, S2 Table), unlike previously evaluated textbooks aimed teaching first aid beginning the grade five in Nepal [36]. Hence, the snakebite first aid and prevention measures should be taught beginning 1st grade through 12th grade (non-medical) students considering improvements in their carelessness and ability of understanding those measures according to their differential needs, mental growth, and cognitive abilities by ages. We suggest to include basic prevention and first aid measures requiring no special skills, training, and supervision to apply them (Box 1.A.a.1–5) in textbooks used by 1st–5th grade students. For the textbooks used by lower secondary level (6th–8th grade) students, we advice to include specific prevention measures against medically highly important, native venomous snakebites [the prevention measures and information about medically relevant snakes in national contexts [70–72]], widely recommended first aid measures requiring no special skills and specific materials, and useless or potentially harmful first aid (Box 1.A.a, 1B). The first aid measures targeted to secondary school level students should be in detail and logical to clarify the need of using suggested first aid measures. To enhance their first aid skills, the demonstrations (as a part of practical courses or competitions) in own class/school can be an effective approach. Considering the mental growth of the secondary level (9th–12th grade) students, their textbooks should include specific prevention measures against all medically important native venomous snakebites [70–72], widely recommended first aid measures including those requiring special skills and specific materials, too (for example, PIB, LCPI, etc.) and useless or potentially harmful measures (Box 1.A.a–A.b, 1B). Next, the university students are future teachers, too, and paramedical students are future healthcare providers at peripheral and/rural healthcare institutions where snakebite patients may approach them initially. Hence, the first aid (and prevention) measures should be taught them in details (with logics of using and not using certain first aid and adequate practical simulations) to enable these paramedical and non-medical university students learning/using first aid (as described in Box 1) and prevention measures [as mentioned in the published sources [70–72]] appropriately when needed. When paramedical personnel deployed in healthcare institution receive a patient with inappropriately applied tourniquet, the ligature should not be removed rapidly. Rather, it should be removed gradually and concomitantly with the antivenom administration to avoid a rush of injected precipitated venom and metabolic toxin resulting in potentially of rapid and fatal respiratory muscle paralysis, leading to respiratory arrest [73]. Therefore, in all snakebite prone nations with inadequate formal education on snakebites, there is a need of periodic evaluation of curriculum texts by selecting school, university, and paramedical textbooks, to highlight the significance of first aid education among children and adults and interviewing key informants such as policy makers, book writers and editors, and CDC's officers to know the reasons of exclusion of "Safety and First Aid" unit from HPE and inconsistencies for teaching snakebite first aid in schools and universities.

## Constraints of this study

- There might be additional books in Nepal that we did not access because there are additional universities, too, which might use similar curriculum as used by the TU and local governments which might develop local curriculum for the formal education on snakebite care at different levels of schools in snakebite prone districts of Nepal. We could not explore these potential sources during this study due to time and financial constraints. However, this highlight of limitation can help to enhance similar further studies. Therefore, our findings cannot be generalized for impact due to materials that we evaluated to entire educational institutions in Nepal.

- Some TSLC (Technical School Leaving Certificate) courses such as ANM (Auxiliary Nursing Midwifery), CMA (Community Medicine Assistant) and Laboratory Assistant courses (e.g., Pre-Diploma in Medical Lab Technology) were phased out in Nepal since 2020 (Article 42, Medical Education Commission Act 2019; https://ctevt.org.np/documents/ctevt-annual-report-2081). The contents of first aid for snakebite were removed from several curriculum texts on which textbooks writing were based. Therefore, books that we analysed previously [36] could not be re-analysed.

- Although the Tribhuvan University is not taking admission in PCL (Proficiency Certificate Level) Nursing since 2020, the CTEVT is running this course. So, we evaluated PCL Nursing courses.

- We could not access published research articles to know what is the situation of the first aid advocated in textbooks in other countries.

## Conclusions

We provide critical insights and need of urgent interventions to strengthen Nepal's formal education by revising snakebite education related textbooks used nationwide. The integration of using properly edited textbooks in formal education system with "Information and Communication Technology" from basic to higher education levels can improve skill and understanding on appropriate use of measures for first aid of snakebites, i.e., use PIB or LCPI effectively with emphasis on avoiding all harmful or useless traditional first aid treatments, and early presentation to the nearest health facilities supplied with antivenoms for effective snakebite management. This can save lives and limbs of populations inhabiting or visiting snakebite prone areas. The persistent teaching of harmful interventions for first aid of snakebites in schools and universities of Nepal suggests a neglect of education sectors and barriers to snakebite management. Additional assessment of the extent of first aid taught in other universities of Nepal to a more precision level and educational campaigns and intervention studies are essential for the elimination of these neglects and barriers. This can help avoiding the teaching or use of harmful and/or useless first aid in snakebite prone areas widely.

## Supporting information

**S1 Table. Nepalese universities, associated institutions, and total students enrollment yearly in Bachelor's Degree in Health/Nursing in which curriculum includes first aid of snakebite.**
(DOCX)

**S2 Table. School and university textbooks currently used in Nepal and having potential of advising measures for first aid of snakebites.**
(DOCX)

**S1 Fig. Interviewing a health education related book writer (left) and three teachers who used to teach first aid practices for snakebites and other accidental illness in university based on Bharatpur Metropolitan City, Chitwan District, Nepal.** Photograph by DPP.
(TIF)

**S2 Fig. Pictorial list of 46 textbooks [with outer/inner pages] reviewed during this study and used for teaching currently in Nepal [textbooks used by school students are displayed in the first four rows and paramedical and university students in the fifth (last) row].** Photograph by DPP.
(TIF)

## Box 1  What to do and what not to do advice if someone is bitten by a venomous snake in Nepal and elsewhere (having diversity of snakes, geo-climates, and socio-economic conditions like in Nepal).

**A] What to do** [recommended advice (RA)]

*A.a. Requiring no special skills and specific materials*

1 Stay **calm** after snakebite (during pre-hospital care and arranging transport, and on the way to the hospital seeking treatment) and **reassure** the snakebite patient. Mostly, the life-threatening venom effects appear only after several hours of snakebite. In an instance of elapid snakebites in Bharatpur, Nepal, the ptosis appeared after 26 hours post-snakebite. The life-threatening symptoms are developed after six to eight hours post-viperid snakebites. Many pitviper bites are rarely life-threatening in Nepal.

2 Move slowly **away from the snake** if snake involved in bite is seen. Carrying victim to a safety place avoids the risks of multiple bites or prevents snakebites to first aiders, too. Never pick up the snake or try to trap or kill it to ensure safety of victims and first aiders. Do not be overreacted with the snake involved in bite. Rather, you can take a photograph which may help in species diagnosis and prognosis of the venom effects on snakebite victim.

3 Remove all **constrictive items** (e.g., rings, jewelleries, etc.) or tight clothing from the bitten limb, other bitten body parts or even around the victim's body to minimize venom effects that cause swelling.

4 **Avoid ambulating** immediately after the bite, even for short distances. Performing any vigorous activities such as walking, running or cycling immediately after snakebite increases the blood circulation predisposing to severe systemic envenoming.

5 Rather, call or ask to call the local helpline number for **ambulance** or arrange other **comfortable transport** even in a suspected snakebite, too. Rural residents engaged in field-based agricultural activities are usually long distances away from home. They may have a challenge for avoiding themselves being ambulated. However, a call for an ambulance or expert help can be a life saving aid.

*A.b. Requiring special skills and specific materials*

1 Know the basic information about antivenom and need of supplying antivenom at a healthcare centre/hospital within one-hour-motor-distance (if possible) from areas at risk of snakebite envenoming.

2 Know healthcare institution provided with **antivenom** and carry victim to this institution within an hour after snakebite. Therefore, seek immediate help for transport of victim to the nearest healthcare facility supplied with antivenom and get medication of envenoming. Also, it is better to know how to swiftly and comfortably transfer a patient to the referral healthcare facility supplied with antivenom. Do not wait for symptoms to appear at the locality of snakebite (because the non-expert assessment to confirm the snakebite by identifying signs of snakebite and cross-checking of observed signs with reported features of snakebite can be the a waste of time). The witness non-expert in snake and snakebite should always consider any snakebite likely to be venomous. Accordingly, victim should be prepared for early access in proper healthcare institution. However, after reaching hospital, tell doctor about evolution of venom effects on the way to hospital. Your monitor of the systemic symptoms' development can be useful to doctors to be prepared for timely in-hospital management of snakebite envenoming.

3 Restrict/reduce the victim's physical activity, particularly the snakebite affected body parts, by any means in order to avoid muscular contraction/movement and slow down the lymphatic flow rate. You can lay the patient on his/her side. **Immobilize** the snake-bitten limb/s using a specific technique for pressure immobilization bandaging (**PIB**) in elapid snakebite causing no local tissue necrosis. Despite the concern of requiring thorough training and potential adverse effects of its improper use, many guidelines widely recommend PIB. If viperid (hemotoxic) snakebite is confirmed, use local compression pad immobilization (**LCPI**). Using PIB and LCPI effectively needs training and special materials to ensure proper tension on the bitten limb and bite site.

4 Provide artificial respiration (AR) or apply chest compression for cardio-pulmonary resuscitation (CPR) if necessary on the way to hospital or on snakebite localities. But, AR/CPR user needs to have related first aid training.

5 If tourniquet was used, it should be gradually released concomitantly with antivenom administration.

6 **Wash the wound** with clean water gently to remove spill over venom if any (particularly in front-fanged elapid (e.g., cobra and krait) snakebite cases) and **examine** the wound and try to confirm snakebite oberving teeth marks on the skin, keep it dry with clean cloth (swab), and mark the bite site properly. Fang marks (i.e., the presence of two puncture wounds) indicate a bite by a venomous snake and several small puncture wounds arranged in an arc are seen suggest a non-venomous snakebite in general.

7 Keep the bitten body part at the level of the victim's heart.

**B] What not to do** [non-recommended advice (NRA)]

1 Do not **incise** and **suck** the snakebite wound to let the injected venom out. The use of mouth or any other suction device is worthless to extract injected venom and causes the local tissue injuries.

2 Do not apply a ligature/**tourniquet** as its use was found to be ineffective or harmful.

3 Do not provide **foods, liquid/water, or stimulants** (e.g., tea/coffee) to the snakebite victim unless it is advised by a doctor.

4 Do not apply **cold compresses** using cooling agent like icepack or cold water bag to the wound as the use of ice causes more harms than benefits.

5 Do not apply any **chemicals, herbals or electric shock** on bite site. Some herbals may have positive effects to lessen swelling and pain. However, effective neutralizing of injected venom by herbals has not been proved yet and not recommended to use or eat any herbals in guidelines for snakebite management .

6 Do not raise the bitten body part above the level of the patient's heart.

## Acknowledgments

We accessed textbooks used in paramedical courses from the libraries of School of Health Sciences, Narayeni Poly-technique Institute, Bharatpur Hospital, and Sri Medical College, Chitwan District and other textbooks from Gandaki and Nobel Boarding Schools, Nawalpur District, Naryeni Model Secondary School, Bharatpur, and Saptagandaki Multiple Campus, Tribhuvan University, Bharatpur, Chitwan. Also, we received several books from bookshops located in Bharatpur Metropolitan City, Chitwan. We would like to thank all libraries and personnel supplying books to scan and printing of the associated contents for the research purpose. Additionally, we sincerely acknowledge Deepak Babu Shrestha, Tika Ram Devkota, and Krishna Prasad Ghimire from Saptagandaki Multiple Campus, Tribhuvan University; Arjun Kumar Baruwal from SOS Balgram School, Bharatpur; and Keshav Sapkota, Rishi Ram Adhikari, and Khimananda Sapkota from Naray-eni Model Secondary School, for the information on currently used textbooks containing "Safety and First Aid". We are also thankful to Dr. Narayan Bahadur Thapa, Chitwan and Sabita Pandey, Nawalpur who supported us in data collection to a certain extent.

## Author contributions

**Conceptualization:** Deb Prasad PANDEY.

**Data curation:** Deb Prasad PANDEY, Bishnu Prasad Khanal, Hardik Sapkota.

**Formal analysis:** Deb Prasad PANDEY.

**Investigation:** Deb Prasad PANDEY, Bishnu Prasad Khanal, Hardik Sapkota.

**Methodology:** Deb Prasad PANDEY.

**Project administration:** Bishnu Prasad Khanal, Hardik Sapkota.

**Supervision:** Deb Prasad PANDEY.

**Validation:** Deb Prasad PANDEY.

**Visualization:** Deb Prasad PANDEY.

**Writing – original draft:** Deb Prasad PANDEY.

**Writing – review & editing:** Deb Prasad PANDEY, Bishnu Prasad Khanal, Hardik Sapkota.

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
