## [Decision Letter · Decision Letter 0]

10 Jun 2025

Analyzing prehospital snakebite management in textbooks used in Nepal: a need of further attention!

Dear Dr.Pandey,

Thank you for submitting your manuscript to PLOS Neglected Tropical Diseases. After careful consideration, we feel that it has merit but does not fully meet PLOS Neglected Tropical Diseases's publication criteria as it currently stands. Therefore, we invite you to submit a revised version of the manuscript that addresses the points raised during the review process.

Please submit your revised manuscript within 60 days. If you will need more time than this to complete your revisions, please reply to this message or contact the journal office at plosntds@plos.org. Please include the following items when submitting your revised manuscript:

We look forward to receiving your revised manuscript.

Kind regards,

Marco Aurélio Sartim, PhD

Guest Editor

José María Gutiérrez

Section Editor

Shaden Kamhawi

co-Editor-in-Chief

Paul Brindley

co-Editor-in-Chief

**Journal Requirements:**

At this stage, the following Authors/Authors require contributions: Deb P. PANDEY, Bishnu Prasad Khanal, and Hardik Sapkota. Please ensure that the full contributions of each author are acknowledged in the "Add/Edit/Remove Authors" section of our submission form.

2) We have noticed that you have uploaded Supporting Information files, but you have not included a complete list of legends. Please add a full list of legends for your Supporting Information files after the references list.

3) Some material included in your submission may be copyrighted. According to PLOSu2019s copyright policy, authors who use figures or other material (e.g., graphics, clipart, maps) from another author or copyright holder must demonstrate or obtain permission to publish this material under the Creative Commons Attribution 4.0 International (CC BY 4.0) License used by PLOS journals. Please closely review the details of PLOSu2019s copyright requirements here: PLOS Licenses and Copyright. If you need to request permissions from a copyright holder, you may use PLOS's Copyright Content Permission form.

Potential Copyright Issues:

- Please confirm (a) that you are the photographer of Figures S1 and S2, or (b) provide written permission from the photographer to publish the photo(s) under our CC BY 4.0 license.

4) In the online submission form, you indicated that "The data that support the findings of this study are available upon request from the corresponding author, DPP.". All PLOS journals now require all data underlying the findings described in their manuscript to be freely available to other researchers, either

- In a public repository

- Within the manuscript itself

- Uploaded as supplementary information.

**Reviewers' Comments:**

Reviewer's Responses to Questions

**Key Review Criteria Required for Acceptance?**

**Methods:**

-Are the objectives of the study clearly articulated with a clear testable hypothesis stated?

-Is the study design appropriate to address the stated objectives?

-Is the population clearly described and appropriate for the hypothesis being tested?

-Is the sample size sufficient to ensure adequate power to address the hypothesis being tested?

-Were correct statistical analysis used to support conclusions?

-Are there concerns about ethical or regulatory requirements being met?

Reviewer #1: need to clearly state how big the data set was, how many pages in each of the 46 textbooks were devoted to snakebite health education.

Line 101: Need to provide definitions of “recommendable advice” and “non recommendable advice”.

Need to inform about the number of textbooks for school students and the number of books for university. Similarly need to inform how many school students were using those books. Such information is necessary to assess/comment on the potential impact at he community level in general.

I suggest adding content analysis rather than a descriptive statistics by quantifying the themes. The dataset is rather small, and the section on first aid is probably a smaller section of a few pages within each book. If descriptive quantitative method is kept, the paper will still benefit from providing some narrative quotes from within the data.

Reviewer #2: The study addresses a pertinent issue regarding the quality of first aid education for snakebite management in Nepal and presents clearly defined objectives. The comparison with a previous study provides valuable context, and the methodological design is appropriate. Nevertheless, the authors are urged to explicitly state the hypothesis in the Introduction, as it is presently only implied. The authors should elucidate the representativeness of the sample in terms of national coverage and adoption, despite the fact that the sample of 46 textbooks appears to be sufficient for a descriptive analysis. The Wilcoxon test and two-proportion Z-test with continuity correction are well-executed and suitable statistical analyses; however, the interpretation of the findings would be further strengthened by the inclusion of measures of effect size and the discussion of the extent of the observed differences. Lastly, the investigation comprises interviews with authors and educators, despite the absence of direct research with human or animal subjects. Consequently, the manuscript should specify whether these informants provided verbal or written assent, even if formal ethical approval was not necessary. By addressing these points, the manuscript's scientific rigor and methodological transparency will be improved.

Reviewer #3: Objectives clearly stated

- Please more clearly describe the 46 textbooks as it pertains to level of education being taught, age groups and under what type of curriculum. Supplemental table 1 is very dense and hard to interpret. In the manuscript we need a simply table to describe this data

- Please describe why these districts, schools and Tribhuvan University were chosen. Are these the most populated regions and where the majority of education is provided in Nepal? How many students are being taught in these regions and how much of the student population in Nepal does it represent?

- I recommend the authors comment on the primary methodology of teaching in these districts and in Nepal. Are most students being taught with physical books? What percentage are being taught via internet or computer resources and does the team know if they are used as online resources for PHCP SBE?

- The sources of guidelines used appear to be Nepal 2019 guidelines and WHO guidelines from 2016.

Reviewer #4: I understand that the main difference between first aid and pre-hospital care (PHC) lies in the depth and specialization of the care provided. First aid is basic and immediate actions that anyone can perform to stabilize a victim before professional help arrives, while PHC is a more specialized care, performed by trained professionals with advanced equipment and protocols. Therefore, I believe it would be better for the authors to use "first aid" throughout the text.

Could the authors clarify how the 7 people arrived to indicate the evaluated teaching materials? - a health education related book writer, five teachers who used to teach PHCPs for snakebites and other accidental illness in university and schools, and an administrator and science teacher of a school. How can the authors ensure that these chosen people are sufficient to make a satisfactory indication of the materials? Is the number of schools visited sufficient?

**Results:**

-Does the analysis presented match the analysis plan?

-Are the results clearly and completely presented?

-Are the figures (Tables, Images) of sufficient quality for clarity?

Reviewer #1: Line 140: Need to inform the reader why quite a few books (11 books that contained no information on safety/first aid and 19 books that did not have snakebite envenoming related information) got identified initially? What was the criteria used to identify.

Line 142: There is a need to inform about the rationale of selecting both school textbooks and university textbooks, highlighting the significance (from community's health education perspective) of grade 6th book relative to the significance of a information included in the curriculum for a bachelor’s degree.

Lines 147-149: Lacks clarity: “for grades 4th, 5th, 8th, 9th, 10th, and B.Ed. 1st–3rd year, recently. Among the currently used textbooks in Nepal, the "Safety and First Aid" unit was included starting the textbooks prescribed for the 4th graders (Supplementary Table 1). But, the advice for snakebite first aid….”

Lines 158-167: some quotes from the reviewed textbooks are needed.

Lines 168-175: comparison with a past study does not add much to the argument. The important is what is the current situation, its reason. If this aspect is to be kept, the previous study needs to be explained in some detail, and the discussion section needs to include some discussion about the factors that led to improvement.

Reviewer #2: The results presented are consistent with the stated analysis plan and hypotheses. The authors effectively compare current textbook content with data from a previous study, using appropriate statistical methods to assess differences in the proportions of recommended and non-recommended prehospital care practices. The presentation of the findings is clear, with relevant metrics such as medians, interquartile ranges, and p-values reported. However, while the use of box plots and tables contributes to understanding the data, the quality of the figures should be verified during typesetting to ensure readability and resolution, particularly for Figures 2 and 3, which play a central role in illustrating improvements and persisting deficiencies. Overall, the results are well-structured and support the conclusions drawn, though the visual materials should be reviewed for clarity and publication standards.

Reviewer #3: Please see comments in Conclusions.

Reviewer #4: Table 1- The text of the book cited as reference (NATIONAL GUIDELINES FOR SNAKEBITE MANAGEMENT IN NEPAL) by the authors does not show it as recommendable advice, see Page 32 – “Pressure immobilization (PIB) is believed to delay in spread of venom to systemic circulation and PIB method is commonly recommended by many experts in pre-hospital management. However, the pressure-immobilization technique demands special equipment and training and is not considered practicable for general use in Nepal. Searching for the material to apply pressure immobilization may cause delay in seeking much needed health care for treatment of envenoming35,36. Moreover, envenoming by cobra and vipers snakes causes local tissue damage and localization of toxin by PIB may worsen tissue damage.”

Na Supplementary Table 1. School and university textbooks currently used in Nepal and having potential of advising measures for prehospital care of snakebites, The books are described and it was possible to identify books from 2010 (15 years ago), are these old books still used? How is the exchange of textbooks carried out in Nepal? Isn't there a book exchange from time to time?

**Conclusions:**

-Are the conclusions supported by the data presented?

-Are the limitations of analysis clearly described?

-Do the authors discuss how these data can be helpful to advance our understanding of the topic under study?

-Is public health relevance addressed?

Reviewer #1: Discussion

Line 180: Lacks clarity: Not clear what is meant by a total of 15 books need immediate re-edition although the advice of harmful, useless or non standard measures in textbooks evaluated during this study were greatly reduced

Line 182-185 includes result not discussion

Most of the discussion is not aligned to the article’s main focus. The article is not about what is a good or incorrect snakebite first aid advice but about what advice is included in textbooks. Therefore, the discussion needs to be around what might be the reasons for incorrect advice being included in the books, what is the situation in other countries/regions. Similarly, the information about websites the context of the websites appears disjointed from the main argument.

Reviewer #2: The conclusions are well supported by the data and highlight the persistence of harmful snakebite first aid advice in textbooks. The authors clearly address the public health relevance and propose meaningful educational interventions. While some limitations are noted, a more explicit discussion of sampling and generalizability would strengthen the interpretation. Overall, the conclusions are valid and relevant to advancing snakebite management education.

Reviewer #3: Conclusions are reasonable and show the need to improve snake envenomation education in these textbooks with updated content. and among these districts.

I recommend to the authors that they clarify the level of education for these textbooks and they future career pathways for the degrees. For example, "medical undergraduate". Is that a medical student who will then acquire a MD and then go on to being labeled a doctor who will then go to a training program for residency?

Are all these textbooks for primary school level and some college level before graduate school or medical school?

I also think the title needs attention to clarify the level of education for the "textbooks used in Nepal" and also clarify if these are the national textbooks for the entire country. This is an important for the readers to understand immediately when analyzing the findings and also government officials who will undoubtedly need to review your findings.

I also recommend a discussion on the snake species which is most commonly encountered from envenomation in Nepal. Speckled cobra, common krait and Russell's viper...

I have concerns about Box 1 and also discussion elements of PIB and LCPI. This is a technique which has shown some efficacy in those bitten by certain Elapids is generally not recommended for Crotalids like the Russell's Viper which is prevalent in Nepal. Author's need to review the Nepal guidelines from 2019 in detail in regards to PIB. " Pressure immobilization (PIB) is believed to delay in spread of venom to systemic circulation and PIB method is commonly recommended by many experts in pre-hospital management. However, the pressure-immobilization technique demands special equipment and training and is not considered practicable for general use in Nepal. Searching for the material to apply pressure immobilization may cause delay in seeking much needed health care for treatment of envenoming35,36. Moreover, envenoming by cobra and vipers snakes causes local tissue damage and localization of toxin by PIB may worsen tissue damage"

- in my opinion this is not a clear recommendation on PIB among Elapids in Nepal and references used for PIB and LCPI, 38, 47-49, are not related Nepalese envenomation. Furthermore, looking at the guidelines from Nepal in 2019 it clearly states that considered practical in Nepal.

- I disagree that the authors make recommendations for using PIB and LCPI in Nepal on lines 247 - 250 and throughout manuscript. Is this the purpose of this investigation? Is this something that should be taught and put into curriculum with unclear evidence? The efficacy of PIB and LCPI is not clear and is not a blanket pre-hospital treatment method for all envenomations in Nepal. It may be beneficial in certain circumstances and with Asian Elapids but with Russell's viper there is one paper from Myanmar among n=15. Application standards and techniques are not well defined and trained professionals are needed to understand how much tension to apply and which materials. Tissue necrosis and delay in care have been expressed by other experts in Australasia and authors mention this is their conclusions.

- I do not see in any of these references that it is recommended to not raise the extremity above the heart. This may be a practice that is reasonable, but also do we have evidence to state we should not do something.

- It is evident that the authors have put a lot of work into reviewing these textbooks but

Reviewer #4: In conclusion the authors wrote “The persistent teaching of harmful interventions for prehospital care of snakebites in schools and universities of Nepal suggests a neglect of education system and barriers to prehospital care of snakebites.” I believe that describing negligence in the educational system is very strong and they should review their conclusions.

**Editorial and Data Presentation Modifications?**

Reviewer #1: I suggest adding content analysis rather than a descriptive statistics by quantifying the themes only. The dataset is rather small, and the section on first aid is probably a smaller section of a few pages within each book. If descriptive quantitative method is kept, the paper will still benefit from providing some narrative quotes from within the data.

Lines 158-167: some quotes from the reviewed textbooks are needed.

Lines 168-175: comparison with a past study (which is not explained to any details) is least useful and does not add much to the argument.

Reviewer #2: The manuscript is generally well written and clearly organized. However, I suggest a careful language revision to improve clarity and reduce redundancy in some sections, particularly the Introduction and Discussion. Some grammatical inconsistencies and overly long sentences may hinder reader comprehension.

Figures and tables are informative, but I recommend ensuring high-resolution versions are provided, especially for Figures 2 and 3, which are central to the study's message. Table captions could also be slightly expanded to ensure clarity without reference to the main text.

Additionally, the “Box 1” summary is a valuable tool for readers and should be carefully formatted for visual clarity. With these minor editorial adjustments, the manuscript will be suitable for publication.

Reviewer #3: (No Response)

Reviewer #4: Minor Revison

**Summary and General Comments**

Reviewer #1: English editing is required

Reviewer #2: The study addresses a relevant and underexplored public health issue with appropriate methodology and sound analysis. The results support the conclusions, and public health implications are clearly stated. Minor revisions are needed to improve hypothesis clarity, discuss sampling limitations more explicitly, enhance language precision, and ensure figure quality. No ethical concerns identified.

Reviewer #3: I commend the authors for their work as it is valuable that our school age children and those training in premedical and paramedical fields know evidence-based pre-hospital practices for snake bite victims. I have concerns with the methods and also with the recommendations for PIB and LCPI. In my opinion, these techniques have not been well studied among Nepalese snake bites and have shown limited evidence of their efficacy. There are real concerns for those without training on appropriate techniques for the placement of PIB and LCPI that they will lead to tissue necrosis and delay in care. It is also unclear why the authors are recommending it's use as the 2019 Nepalese guidelines are not clear about recommending this practice and frankly say it is "not practical for general use in Nepal". My thoughts align with this recommendation that it is not for general use, needs specialized training and likely will only benefit certain snake bite victims. Unfortunately, I have to recommendation rejection at this time.

Reviewer #4: The topic addressed is interesting and pertinent. Initial care in snakebite accidents is rarely addressed and the use of educational materials seems to be a support point to prepare young people for this emergency care. Some details and adjustments are necessary to better understand the research as listed in the other items of the report.

PLOS authors have the option to publish the peer review history of their article (what does this mean? ). If published, this will include your full peer review and any attached files.

**Do you want your identity to be public for this peer review?** For information about this choice, including consent withdrawal, please see our Privacy Policy .

Reviewer #1: No

Reviewer #2: No

Reviewer #3: No

Reviewer #4: No

**Figure resubmission:**

**Reproducibility:**



---

## [Decision Letter · Decision Letter 1]

5 Aug 2025

Analyzing first aid in textbooks used by non-medical and paramedical students in Nepal: a need of further attention for snakebite management!

Dear Dr. Pandey,

Thank you for submitting your manuscript to PLOS Neglected Tropical Diseases. After careful consideration, we feel that it has merit but does not fully meet PLOS Neglected Tropical Diseases's publication criteria as it currently stands. Therefore, we invite you to submit a revised version of the manuscript that addresses the points raised during the review process.

Please submit your revised manuscript within 30 days Sep 04 2025 11:59PM. If you will need more time than this to complete your revisions, please reply to this message or contact the journal office at plosntds@plos.org. Please include the following items when submitting your revised manuscript:

* A rebuttal letter that responds to each point raised by the editor and reviewer(s). You should upload this letter as a separate file labeled 'Response to Reviewers '. This file does not need to include responses to any formatting updates and technical items listed in the 'Journal Requirements' section below.

* A marked-up copy of your manuscript that highlights changes made to the original version. You should upload this as a separate file labeled 'Revised Manuscript with Track Changes '.

* An unmarked version of your revised paper without tracked changes. You should upload this as a separate file labeled 'Manuscript '.

We look forward to receiving your revised manuscript.

Kind regards,

Marco Aurélio Sartim, PhD

Academic Editor

Section Editor

Shaden Kamhawi

co-Editor-in-Chief

Paul Brindley

co-Editor-in-Chief

**Journal Requirements:**

At this stage, the following Authors/Authors require contributions: Deb P. PANDEY, Bishnu Prasad Khanal, and Hardik Sapkota. Please ensure that the full contributions of each author are acknowledged in the "Add/Edit/Remove Authors" section of our submission form.

**Reviewers' comments:**

Reviewer's Responses to Questions

**Key Review Criteria Required for Acceptance?**

**Methods**

-Are the objectives of the study clearly articulated with a clear testable hypothesis stated?

-Is the study design appropriate to address the stated objectives?

-Is the population clearly described and appropriate for the hypothesis being tested?

-Is the sample size sufficient to ensure adequate power to address the hypothesis being tested?

-Were correct statistical analysis used to support conclusions?

-Are there concerns about ethical or regulatory requirements being met?

Reviewer #1: Method is described well.

Reviewer #3: Authors have provided clarifications for Methods as requested by peer reviewers. I find the changes made to be sufficient. No additional comments

Reviewer #5: yes, addressed reviewer comments.

**Results**

-Does the analysis presented match the analysis plan?

-Are the results clearly and completely presented?

-Are the figures (Tables, Images) of sufficient quality for clarity?

Reviewer #1: At the first review, it was suggested that a distinction be made between what ought to be included in a textbook for 4th/6th/7th grade students compared to the information that ought to be included in a text book for a bachelor degree about health. For example, information about AV, Antivenom availability at a health centre, how to transfer a patient to the facility may not be needed for a 4th or 7th grade student. Hence, evaluation of textbook may only be with reference to what ought to be / is appropriate for a particular level of schooling. For the same reason, judging a textbook whether it has complete or partial information needs to be contextualised with reference to what is the appropriate level of information for that grade. Results still lack clarity in this regard.

Reviewer #3: Authors have made several changes and added material for the Results sections as indicated by the peer reviewers and academic editor. Some limitations are discussed which are acknowledged by the author team. Results of the study will shine light on T. cruzi infection and variability within this region of Brazil. Mixed infection among the cohort is interesting. The diversity of DTUs within this region is also important.

Reviewer #5: Corrected accordingly.

**Conclusions**

-Are the conclusions supported by the data presented?

-Are the limitations of analysis clearly described?

-Do the authors discuss how these data can be helpful to advance our understanding of the topic under study?

-Is public health relevance addressed?

Reviewer #1: Discussion section is rather too long. The research was not about what is a correct or incorrect method of first aid. Hence, all the detailed write up about the accurate/useful/harmful method of first aid could be removed from the discussion section. The discussion needs to focus on why/why not textbook contain/not contain health related information and why incorrect information gets included, what are the ways to manage to that etc. Additionally, results are repeated in the discussion. Such repeat description of results needs to be removed.

Reviewer #3: Conclusions are reasonable. Limitations are also mentioned. One correction that needs to be made is the citation found on line 341. it is the wrong reference to T. melanica. Should be: "Valença-Barbosa C, Finamore-Araujo P, Moreira OC, Alvarez MVN, Borges-Veloso A, Barbosa SE, Diotaiuti L, de Souza RCM. High Parasitic Loads Quantified in Sylvatic Triatoma melanica, a Chagas Disease Vector. Pathogens. 2022 Dec 8;11(12):1498. doi: 10.3390/pathogens11121498. PMID: 36558833"

Reviewer #5: yes.

**Editorial and Data Presentation Modifications?**

Reviewer #1: See the comments about the result section i.e how to [analyze] present the results with reference to grade/level of education

Reviewer #3: Accept

Reviewer #5: yes.

**Summary and General Comments**

Reviewer #1: Overall, the comments and issues raised at the first review are addressed well.

Reviewer #3: Authors have sufficiently addressed all the suggestions and clarifications provided by the peer reviewers and academic editor. Study has validity and will add to the medical literature for this region of Brazil and the complexity of T. cruzi infection in humans.

Reviewer #5: yes.

PLOS authors have the option to publish the peer review history of their article (what does this mean? ). If published, this will include your full peer review and any attached files.

**Do you want your identity to be public for this peer review?** For information about this choice, including consent withdrawal, please see our Privacy Policy .

Reviewer #1: No

Reviewer #3: **Yes: ** Norman L. Beatty, University of Florida College of Medicine

Reviewer #5: No

**Figure resubmission:**
---

## [Decision Letter · Decision Letter 2]

24 Sep 2025

Response to Reviewers
Revised Manuscript with Track Changes
Manuscript

Shaden Kamhawi

co-Editor-in-Chief

Paul Brindley

co-Editor-in-Chief

**Editor Comments:**

**Comments to the Authors:**

**Please note that one review is uploaded as an attachment.**

**Reviewers' comments:**

**Key Review Criteria Required for Acceptance?**

**Methods**

-Are the objectives of the study clearly articulated with a clear testable hypothesis stated?

-Is the study design appropriate to address the stated objectives?

-Is the population clearly described and appropriate for the hypothesis being tested?

-Is the sample size sufficient to ensure adequate power to address the hypothesis being tested?

-Were correct statistical analysis used to support conclusions?

-Are there concerns about ethical or regulatory requirements being met?

Reviewer #1: Overall, good description of methods

The information about schools and universities provided in the method section does not belong here. It could be placed at the end of the introduction section as part of the 'rationale' [potential significance of this research]

Reviewer #5: Given comments addressed.

**Results**

-Does the analysis presented match the analysis plan?

-Are the results clearly and completely presented?

-Are the figures (Tables, Images) of sufficient quality for clarity?

Reviewer #1: Results and analysis align with the objective of the research.

Reviewer #5: Given comments addressed.

**Conclusions**

-Are the conclusions supported by the data presented?

-Are the limitations of analysis clearly described?

-Do the authors discuss how these data can be helpful to advance our understanding of the topic under study?

-Is public health relevance addressed?

Reviewer #1: The first paragraph of the discussion is rather a 'recommendation'. This needs to be moved towards the end of the discussion section.

A recommendation states that first aid education should be included in the books for the first grade. This recommendation needs to be based on evidence (reference) of effectiveness of public health prevention / first aid methods for snakebite or other similar public health issues for children at first grade level.

Discussion still includes lengthy description of first aid methods. That description is not relevant and is outside the scope of this work. The reader could be referred to WHO guidelines to learn about and include in their books the correct methods. Instead, the discussion section should include a para or two about the possible reasons to authors including incorrect information or inadequate information in their books.

Reviewer #5: Not given any comments.

**Editorial and Data Presentation Modifications?**

Reviewer #1: None

Reviewer #5: Accept.

**Summary and General Comments**

Reviewer #1: The authors have adequately addressed most of the issues highlighted at the first review.

Reviewer #5: Given comments addressed and improved.

PLOS authors have the option to publish the peer review history of their article (what does this mean? ). If published, this will include your full peer review and any attached files.

**Do you want your identity to be public for this peer review?** For information about this choice, including consent withdrawal, please see our Privacy Policy .

Reviewer #1: No

Reviewer #5: No

**Figure resubmission:**

**Reproducibility:**

To enhance the reproducibility of your results, we recommend that authors of applicable studies deposit laboratory protocols in protocols.io, where a protocol can be assigned its own identifier (DOI) such that it can be cited independently in the future. Additionally, PLOS ONE offers an option to publish peer-reviewed clinical study protocols. Read more information on sharing protocols at https://plos.org/protocols?utm_medium=editorial-email&utm_source=authorletters&utm_campaign=protocols

---

## [Decision Letter · Decision Letter 3]

17 Nov 2025

Dear Dr Pandey,

We are pleased to inform you that your manuscript 'Analyzing first aid in textbooks used by non-medical and paramedical students in Nepal: a need of further attention for snakebite management!' has been provisionally accepted for publication in PLOS Neglected Tropical Diseases.

Best regards,

Marco Aurélio Sartim, PhD

Academic Editor

José María Gutiérrez

Section Editor

Shaden Kamhawi

co-Editor-in-Chief

Paul Brindley

co-Editor-in-Chief

Reviewer's Responses to Questions

**Key Review Criteria Required for Acceptance?**

**Methods**

-Are the objectives of the study clearly articulated with a clear testable hypothesis stated?

-Is the study design appropriate to address the stated objectives?

-Is the population clearly described and appropriate for the hypothesis being tested?

-Is the sample size sufficient to ensure adequate power to address the hypothesis being tested?

-Were correct statistical analysis used to support conclusions?

-Are there concerns about ethical or regulatory requirements being met?

Reviewer #1: Methods section was reviewed and improved as per suggestion during the previous round of review.

Reviewer #5: No new comments.

**Results**

-Does the analysis presented match the analysis plan?

-Are the results clearly and completely presented?

-Are the figures (Tables, Images) of sufficient quality for clarity?

Reviewer #1: Results section reads well now. The recommendations are taken out and presented in the discussion section

Reviewer #5: No new comments.

**Conclusions**

-Are the conclusions supported by the data presented?

-Are the limitations of analysis clearly described?

-Do the authors discuss how these data can be helpful to advance our understanding of the topic under study?

-Is public health relevance addressed?

Reviewer #1: Discussion is improved; however, it still contains a lot of information on 'what method is correct or incorrect'. That takes the focus away from the aim of this paper i.e. whether textbooks include incorrect information, and why".

Reviewer #5: Satisfactory.

**Editorial and Data Presentation Modifications?**

Reviewer #1: minor edits

Page 13 260-264: The two statements seem contradictory to each other

Page 13 – 270-272. not clear what is meant by ‘ought to”; probably authors intend to say that there was no difference in content (first aid measure) taught to primary School children and university degree students ?

Page 14: 296. Need to remove the word ‘ineffective’

Reviewer #5: Accepted.

**Summary and General Comments**

Reviewer #1: The paper is much improved. It would benefit from further edit or the discussion section.

Reviewer #5: Satisfactory.

PLOS authors have the option to publish the peer review history of their article (what does this mean? ). If published, this will include your full peer review and any attached files.

**Do you want your identity to be public for this peer review?** For information about this choice, including consent withdrawal, please see our Privacy Policy .

Reviewer #1: No

Reviewer #5: No

---

## [Editor Report · Acceptance letter]

Dear Dr. PANDEY,

We are delighted to inform you that your manuscript, "Analyzing first aid in textbooks used by non-medical and paramedical students in Nepal: a need of further attention for snakebite management!," has been formally accepted for publication in PLOS Neglected Tropical Diseases.

Best regards,

Shaden Kamhawi

co-Editor-in-Chief

Paul Brindley

co-Editor-in-Chief
